# Bioengineered Ciprofloxacin-Loaded Chitosan Nanoparticles for the Treatment of Bovine Mastitis

**DOI:** 10.3390/biomedicines10123282

**Published:** 2022-12-19

**Authors:** Preeti Yadav, Awadh Bihari Yadav, Preksha Gaur, Vartika Mishra, Zul-I Huma, Neelesh Sharma, Young-Ok Son

**Affiliations:** 1Centre of Biotechnology, University of Allahabad, Pryagraj 211002, India; 2Division of Veterinary Medicine, Faculty of Veterinary Sciences and Animal Husbandry, Sher-e-Kashmir University of Agricultural Sciences and Technology of Jammu, R.S. Pura, Jammu 181102, India; 3Department of Animal Biotechnology, Faculty of Biotechnology, College of Applied Life Sciences, Interdisciplinary Graduate Program in Advanced Convergence Technology and Science, Jeju National University, Jeju 690756, Republic of Korea

**Keywords:** mastitis, ciprofloxacin, nanoparticles, drug delivery

## Abstract

Mastitis is the most devastating economic disease in dairy cattle. Mastitis in dairy cattle frequently occurs during the dry period or during early lactation. *Escherichia coli* (*E. coli*) and *Staphylococcus aureus* (*S. aureus*)are the main causative agents of mastitis in India. *S. aureus* can form microabscesses in the udder and develop a subclinical form of mastitis. This bacterial property hinders an effective cure during the lactation period. Antimicrobials used for treatments have a short half-life at the site of action because of frequent milking; thereforethey are unable to maintain the desired drug concentration for effective clearance of bacteria. We demonstrated the potential of ciprofloxacin-encapsulated nanocarriersthat can improve the availability of drugs and provide an effective means for mastitis treatment. These drug-loaded nanoparticles show low toxicity and slow clearance from the site of action. Antimicrobial activity against clinical strains of *E. coli* and *S. aureus* showed that the zone of inhibition depended on the dose (0.5 mg to 2 mg/mL nanoparticle solution from 11.6 to 14.5 mm and 15 to 18 mm). These nanoparticles showed good antimicrobial activity in broth culture and agar diffusion assay against bacteria.

## 1. Introduction

Mastitis is economically the most important disease in dairy cattle. It reduces total milk production from 5–25% and in extreme cases 83% [1,2]. The economic losses due to mastitis in the United States (US) are 400–600 million USD, while in India cases have increased about 115 fold in the last five decades, and presently they amount to 995 million USD per annum [1,2,3]. Intramammary infection in dairy cattle frequently occurs during the early dry period or early lactation. Most mastitis cases are of bacterial origin, and just a few species of bacteria account for most cases, such as *Escherichia coli*, *Staphylococcus aureus*, *Streptococcus uberis*, *Streptococcus dysgalactiae*, *Streptococcus agalactiae*, *Streptococcus bovis*, and *Klebsiella pneumoniae* [4,5,6]. *Staphylococcus aureus,* the main bacterium causing mastitis in dairy cows, has the ability to form microabscesses in the mammary parenchyma of the udder and develop a subclinical form of mastitis. This encapsulation property hinders bacteriological cure during lactation because antimicrobials used in this phase have a short half-life in the target of action due to frequent milking and are therefore unable to maintain therapeutic levels long enough to determine the complete elimination of bacteria from their hiding place. Another challenge in the treatment of *S. aureus* infection is its ability to form biofilms; bacteria in biofilms are more resistant to antimicrobials compared to planktonic cells [7]. They are tightly packed in an extracellular polysaccharide matrix, which helps them escape the immune response and antimicrobials in the environment [8]. This matrix hinders the penetration of many antimicrobials, resulting in a significant decrease in antimicrobial efficacy. In addition, bacterial cells in deep layers of biofilms have a slow rate of metabolism and growth due to limited nutrient access [7]. NP-based drug delivery systems introduce a wide range of therapeutics by either binding the drug to their large surface area or carrying it to the site of infection effectively, safely, and with a controlled rate of targeted delivery. They are capable of disrupting bacterial membranes and hindering biofilm formation, thus reducing the survival of the microorganism. A study by Thomas et al. evaluated the efficacy of sustained-release PLGA micro- and nanoparticles containing ciprofloxacin against bacterial biofilms and demonstrated a sustained release over 6 days to effectively eradicate culturable *S. aureus*. The formulation might be a valuable alternative for the treatment of biofilms by achieving high local and sustained drug concentrations while minimizing adverse systemic effects and improving patient compliance [9].

Nanocarriers have the ability to provide controlled drug release over a long time to maintain the desired level of drug at the infection site. The last few decades have seen unprecedented use of nanoparticles for drug delivery, and various drug delivery platforms, especially liposomes, polymeric nanoparticles, dendrimers, and inorganic nanoparticles, have received significant attention. Drugs’ interaction with nanocarriers through physical encapsulation, adsorption, or chemical conjugation has exhibited improvement in their pharmacokinetics profiles and therapeutic indexes. Persistent research efforts and encouraging results have resulted in numerous nanoparticle-based drug delivery systems being approved for the clinical treatment of infectious diseases, while many others are currently under various stages of preclinical and clinical studies [10,11]. Despite the presence of various preventive measures and management practices, there is an immediate need for effective therapeutics to treat mastitis. Chitosan is one of the most commonly used natural polymers in nanomedicine production because it displays desirable characteristics for drug delivery and has been proven very effective when formulated in a nanoparticulate form. Properties such as its cationic character and solubility in an aqueous medium have been reported as determinants of the success of this polysaccharide [12]. However, its most attractive property is the ability to adhere to mucosal surfaces, leading to a prolonged residence time at drug absorption sites and enabling higher drug penetration [12]. Chitosan has further demonstrated the capacity to enhance macromolecules’ epithelial permeation through the transient opening of tight epithelial junctions [12]. In addition, the polymer is known to be biocompatible and exhibit very low toxicity toward mammalian cells, two mandatory requisites for drug delivery applications [12]. Noticeably, chitosan has been described as more efficient at enhancing drug uptake when formulated in a nanoparticulate form than in a solution [12].

Few studies from India, Pakistan, China, and other parts of the world have demonstrated the high efficacy of ciprofloxacin against mastitis [12]. Ciprofloxacin (CPX) is the third generation of fluoroquinolone-based antimicrobials that show antimicrobial activity against Gram-positive and Gram-negative bacteria, and its effective inhibition concentration is very low as reported by different groups [5,13,14].

Ananoparticulate formulation administration by intramammary route is a promising alternative for the effective treatment of bovine mastitis. Therefore, the present study was planned to develop an intramammary nanoparticulate drug delivery system using chitosan (CS) as the polymer due to its muco-adhesive and biodegradable properties and ciprofloxacin as the model therapeutic drug because of its efficacy against *S. aureus*.

## 2. Materials and Methods

### 2.1. Materials

Ciprofloxacin was obtained from Himedia, Mumbai, (India), chitosan, sodium triphosphate pentabasic (TPP), fluorescein isothiocyanate (FITC) dye, and phorbol myristate acetate (PMA). Glacial acetic acid; NaOH; dimethyl sulphoxide(DMSO); trehalose; phosphate buffer saline (PBS); syringe filter (0.22 μ pore size);and RPMI culture media were obtained from Himedia, Mumbai (India).

### 2.2. Nanoparticle Preparation

Drug-loaded nanoparticles were prepared by adding different amounts of ciprofloxacin hydrochloride (CPX) (0.5, 1.0, and 1.5 mg/mL) into the chitosan solution, and the below-mentioned method was used. FITC dye-loaded chitosan nanoparticles were prepared by adding 200 μL of FITC-DMSO (1 mg/mL) solution to the chitosan solution and polymerized with TPP in the dark to avoid fluorescence quenching of the dye. The chitosan nanoparticles were prepared using the ionic gelation method [15]. Briefly, chitosan (1–2% *w*/*v*) was dissolved in 1% (*v*/*v*) glacial acetic acid solution and the pH adjusted to 5 using 1 N NaOH. Then, TPP solution (0.5–1 mg/mL; adjusted to pH 5) was added to the chitosan solution and the mixture was stirred (750 rpm) for 30 min at room temperature.

#### 2.2.1. Nanoparticle Size Determination

Nanoparticle size measurement was performed using a Zetasizer Nano ZS90 (Malvern Instrument, Westborough, MA, USA). Nanoparticles were resuspended in water and transferred to a disposable cuvette. A sample-containing cuvette was placed into the equipment, and three measurements were taken to record the size of the nanoparticles at 25 °C.

#### 2.2.2. Nanoparticle Zeta Potential

The surface charge on the nanoparticles was measured using a Zetasizer ZS90 (Malvern Instrument, Westborough, MA, USA). In brief, a 50 μL chitosan nanoparticle sample was diluted 20 times and transferred to a specialized cuvette for zeta potential measurement. A specialized cuvette containing samples was placed into the sample holder, and three measurements were taken at 25 °C.

#### 2.2.3. Surface Morphology Study

The surface morphology of the CPX-loaded nanoparticles was characterized using a ZEISS scanning electron microscope (Tescan USA Inc., Alameda, CA, USA). Lyophilized nanoparticles with trehalose powders were mounted onto aluminum stubs using double-sided adhesive tape. Samples were then made electrically conductive by coating with a thin layer of gold in a vacuum using a Polaron SC500 gold sputter coater (QuotumTechnologies, Newhaven, UK). The coated specimen was then examined under the microscope at an acceleration voltage of 10 kV.

### 2.3. Encapsulation Efficiency

The encapsulation efficiency of the CPX-loaded nanoparticles was determined using an indirect method [16]. The encapsulation efficiency and loading capacity of the nanoparticles were determined after collection of the nanoparticles by centrifugation at 12,000 rpm for 70 min at 4 °C. The amount of free CPX in the supernatant was measured spectrophotometrically (Thermo Scientific) at 276 nm. The encapsulation efficiency (EE%) and the loading capacity (LC%) of the nanoparticles were calculated as follows:$$\mathrm{EE}\;\left(\%\right)\frac{\mathrm{Total}\;\mathrm{ciprofloxacin}\;\mathrm{or}\;\mathrm{isoniazid}\;-\;\mathrm{free}\;\mathrm{ciprofloxacin}\;\mathrm{or}\;\mathrm{isoniazid}\;\mathrm{in}\;\mathrm{supernatant}}{\mathrm{ciprofloxacin}\;\mathrm{or}\;\mathrm{isoniazid}\;\mathrm{taken}}\times 100$$
$$\mathrm{LC}\;\left(\%\right)=\frac{\mathrm{Total}\;\mathrm{ciprofloxacin}\;\mathrm{or}\;\mathrm{isoniazid}-\;\mathrm{free}\;\mathrm{ciprofloxacin}\;\mathrm{or}\;\mathrm{isoniazid}\;\mathrm{in}\;\mathrm{supernatant}}{\mathrm{Weight}\;\mathrm{of}\;\mathrm{nanoparticles}}\times 100$$

### 2.4. Release Study

Ciprofloxacin release from nanoparticles was studied using a dialysis method. In brief, the CPX-loaded chitosan nanoparticles were collected after centrifugation at 12,000 rpm for 70 min. The nanoparticles were transferred to a dialysis bag (containing the 5 mg equivalent of free CPX) with a molecular weight cutoff of 12 kDa, and the dialysis bag was immersed in 100 mL phosphate-buffered saline at pH 7.4 or pH 5.2 (to simulate body fluid pH and macrophage interior pH, respectively) at 37 °C under continuous shaking. At a predetermined time, 4 mL of dialytic medium was taken out for analysis and an equal volume of freshly prepared PBS was added. The absorbance of the collected sample was taken at 276 nm by a UV-vis spectrometer. The released amount of CPX was quantified by referring to a calibration curve recorded from known amounts of CPX at the same condition.

### 2.5. Uptake Study

A differentiated THP-1 cell line was used in the uptake study. In brief, THP-1 cells were seeded in a 6-well plate (0.1 × 106 cells/well) with 10% FBS RPMI media and 20 nM PMA and incubated at 37 °C for 36 h in a CO_2_ incubator (5% CO_2_ and 95% RH). After 36 h, the supernatant was removed and fresh media was added to each well. The differentiated THP-1 cells were then incubated with FITC alone and FITC dye-loaded nanoparticles for 6 h. Each well was washed three times with PBS. Cells were scraped using a scraper and cell suspension was analyzed for nanoparticle uptake study using a flow cytometer (BD FACS Calibur, Bergen, NJ, USA).

### 2.6. Toxicity Assays

Nanoparticle-induced cytotoxicity study was performed using different assays such as the trypan blue dye exclusion assay, neutral dye uptake and accumulation assay, and hemoglobin release assay on the different cells (3T3 fibroblast cell lines and RBCs).

#### 2.6.1. Trypan Blue Dye Exclusion Test

The trypan blue dye exclusion test was performed according to the method described by Tennant [17]. Failure to exclude trypan blue reflects a loss of plasma membrane integrity associated with necrosis [18]. Three replicates per concentration were maintained. Cells were mixed with equal volumes of trypan blue stain, observed under a microscope, and counted for stained cells.

#### 2.6.2. Neutral Red Uptake Assay

The neutral red (NR) uptake assay was performed to determine the accumulation of neutral red dye in the lysosomes of viable cells [19]. The uptake of NR into the lysosomes/endosomes and vacuoles of living cells is used as a quantitative indication of cell number and viability. The neutral red retention assay was performed according to the methods described by Guillermo et al. [20]. In brief, 3T3 fibroblast cell lines were seeded into 96-well plates (10,000 cells per well) and incubated overnight for cell recovery and adherence and exponential growth. Cells were treated with different concentrations (1, 10, 100, or 10,000 μg/mL) of CS-NPs, CPX-CS NPs, or vehicle (PBS-buffer, pH 7.4) alone as a negative control for 24 h at 37 °C under a humidified atmosphere of 5% CO_2_. All experiments were performed in triplicate. The cell viability (%) relative to the control wells containing cell culture medium without nanoparticles or PBS as the following equation calculated a vehicle:Cell Viability = OD-test/OD-control × 100
where OD-test is the light absorbance (540 nm) of the test sample and OD-control is the light absorbance (540 nm) of the control sample. Each sample was measured in triplicate from three independent experiments.

#### 2.6.3. Hemoglobin Release Assay

Blood compatibility studies were performed with RBCs isolated from a goat blood sample. In brief, 10 mL of goat blood sample was obtained from a slaughterhouse in a tube containing anticoagulant EDTA after proper mixing and centrifuged at 1500× *g* for 10 min. Red blood cells (RBCs) were collected and washed thrice with phosphate buffer solution (PBS) and spun for 7 min at 1000× *g*. RBCs were resuspended in PBS and diluted up to 20% with erythrocyte stock solution. The nanoparticle suspension was added to the erythrocyte stock solution at different concentrations (1, 10, 100, or 1000 μg/mL) of CS-NPs, CPX-CS NPs, or vehicle (normal saline) alone as a negative control for 2 h. The mixtures were incubated at 37 °C in a continuous shaking water bath for 2 h. After centrifugation at 1000× *g* for 5 min, the supernatant absorbance was recorded at 560 nm. The saline solution alone was used as the negative control (0% lysis) and 0.1% triton in PBS was used as a positive control (100% lysis) [21]. The percent of hemolysis was calculated using the formula:Hemolysis (%) = (OD sample − OD negative control) × 100OD positive control − OD negative control

### 2.7. Lipid Peroxidation Assay

Membrane damage in RBCs was determined by quantifying the release of malondialdehyde (MDA) after nanoparticle exposure. MDA is a product of lipid peroxidation, and its release levels indicate membrane damage. Lipid peroxidation in erythrocytes was assayed with a method described by Stern et al. [22]. In brief, RBCs were incubated with CS NPs, CPX-CS NPs, or CPX in equivalent amounts of drug in solution with 1 mL of RBCs(20%); 1 mL of 10% ice-cold trichloroacetic acid was added and vortexed. The mixture was centrifuged at 3000 rpm for 10 min followed by the addition of 1 mL 0.67% *w*/*v* 2-thiobarbituric acid (in 0.1 N NaOH) to 1 mL supernatant, which was kept in a boiling water bath for 10 min. The final reaction mixture was cooled, diluted with 1 mL distilled water, and absorbance was recorded at 535 nm. The results were expressed as nM MDA formed/mL erythrocytes. The molar extinction coefficient of the MDA-TBA complex at 535 nm was 1.56 × 108/M/cm.
Lipid peroxidation = (OD × total volume of reaction mixture) × 10^9^ × 10EC—amount of sample taken

### 2.8. Minimum Inhibitory Concentration Determination (MIC) of CPX-CS NPs 

The minimum inhibitory concentration and antimicrobial activity of CPX in solution and in CPX NPs were determined using a microdilution test in a culture broth by comparing samples of free ciprofloxacin to samples of nanoparticles containing equivalent amounts of the drug. The bacterial inoculum was prepared from a Mueller–Hinton plate that had been streaked with a single colony from an initial subculture plate and incubated for 18 to 24 h. The culture growth was carried out in 96 well ELISA plates after the inoculum was adjusted to 10^5^ CFU/mL in Mueller–Hinton liquid medium in a total volume of 100 μL per well. The cultures were treated with 8, 4, 2, 1, 0.5, 0.25, 0.125, and 0.0625 μg/mL (8 levels) calculated by the concentration of CPX, and 100 μL from each level was added into a 96-well plate. In addition to the control, we used chitosan nanoparticles without the drug, i.e., blank nanoparticles (CS-NPs) at 100 μg/mL in Mueller–Hinton liquid medium. Mueller–Hinton liquid medium without CPX was also used to inoculate bacteria using the same method as the blank control. The sample was incubated at 37 °C and removed after 24 h, then the lowest concentration of no bacterial growth was determined as the minimum inhibitory concentration (MIC). A positive control (growth) formed by a culture broth containing micro-organisms, a negative control (sterility) formed by a culture broth without micro-organisms, and a drug control consisting of a culture broth containing the highest ciprofloxacin concentration were included.

### 2.9. Antibacterial Activity

The antibacterial activity of CPX-loaded chitosan nanoparticles was studied using the agar diffusion assay method. In brief, CPX-CS NPs containing 0.15 μg/mL equivalent of free CPX were used in this study. For the antimicrobial activity of the formulations, major mastitis-causing bacteria such as *Staphylococcus aureus* (*S. aureus*) and *Escherichia coli* (*E. Coli*) were collected from clinical mastitis milk samples from previous research [4,23] and used in the present in vitro study. A molten Mueller–Hinton agar stabilized at 45 °C was seeded with 0.1 mL of a 24 h broth culture of the test organisms(*S. aureus* and *E. coli*) containing approximately 10^5^ cfu/mL in a sterile petri dish and allowed to set. Wells of 6 mm diameter were created with a sterile cork borer and filled with 50 µL of nanoparticles or free drug. The plates were pre-incubated for 1 h at room temperature to allow for diffusion of the solution and then incubated for 24 h. The zones of inhibition were measured (mean, *n* = 3).

## 3. Results

### 3.1. Ciprofloxacin-Loaded Nanoparticle Characteristics

Ciprofloxacin-loaded chitosan nanoparticles were prepared using the gelation method by adding TTP. The CS/TPP weight ratio influenced the particle size and zeta potential of the nanoparticles; by increasing the CS/TPP weight ratio, nanoparticle size and zeta potential were increased (195.6 nm to 229.1 nm and +24.86 to +28.35 mV). The colloidal stability of the prepared formulations was observed after 1month of storage in the refrigerator. Particle aggregation was measured in terms of increase in polydispersity; different size of the same particles at different times was observed in one formulation of CS/TPP with a weight ratio of 9:1 and not in the 5:1 ratio. Therefore, the formulation with a 5:1 weight ratio was selected for further study (300.8 nm and PDI was 0.350). An increase in chitosan concentration from 1 to 2% *w*/*v* affected the size and zeta potential of the optimized nanoparticles, as shown in Table 1. An increase in CPX concentration led to a decrease in encapsulation efficiency. Nevertheless, CPX was entrapped in the matrix of nanoparticles to an appreciable extent (43–47%).

CPX-loaded NP surface morphology was studied after lyophilization, and trehalose was used as a cryoprotectant. The surface morphology study revealedthat the size of particles was around 250 nm and were irregular and sticky in shape. Particles were deposited on the trehalose iceberg (Figure 1).

### 3.2. Drug Release and Stability of Chitosan Nanoparticles

The release profiles of the CPX-loaded chitosan nanoparticles are shown in Figure 2, where ciprofloxacin in solution was released rapidly from the dialysis bag within the first 12 h. The CPX-loaded chitosan nanoparticles showed a biphasic pattern with an initial-burst drug release followed by a more sustained release, with about 1% of the drug immediately released in 1 h and 27% in 24 h in pH 7.4 release medium. Similarly, at pH 5.2, the CPX-loaded nanoparticles immediately released about 1% of the drug in 1 h and 13% in 24 h.

The stability of nanoparticles prepared by gelation method is an essential requirement for pharmaceutical applications, as the storage stability and biocompatibility of the nanoparticles is a great concern. It is known that tiny particles are inclined to agglomerate with each other to reduce their surface area, and hence reduce their free surface energy. The stability of the prepared nanoparticles was analyzed at 4 °C and room temperature for 30 days. All the particles with a 5:1 weight ratio were rather stable, with negligible size fluctuation of 201.6 nm and PDI-0.269, while particles with a 9:1 weight ratio showed size increment and decrease in drug content (Table 2).

### 3.3. Cellular Uptake of FITC-Loaded Chitosan Nanoparticles

Bacteria reside in cells of the mammary gland; it is therefore very crucial for nanoparticles to enter into cells for efficient killing of bacteria inside affected cells. Uptake of chitosan nanoparticles was studied in differentiated THP-1 cells by flow cytometry study (Figure 3). The uptake of FITC-loaded nanoparticles by differentiated THP-1 cells was measured by flow cytometry; 10,000 cell events for each sample were acquired. Unlabeled and labelled cells were used as positive and negative controls to set the voltage of the detector for this experiment. In this study, it was clearly shown thatFITC-loaded NPs were maximally taken by differentiated THP-1 cells.

### 3.4. Antimicrobial Activity of Nanoparticles against Clinical Isolates 

The antimicrobial activity of CPX-loaded nanoparticles was evaluated against clinical isolates of *E. coli* and *S. aureus* with an agar diffusion study (Figure 4). The efficacy of the antimicrobial activity of CPX-loaded nanoparticles is dose dependent, higher amounts of nanoparticles show a high zone of inhibition on an agar plate. It was also observed that antimicrobial activity against *S. aureus* was more efficient in comparison to *E. coli*.

### 3.5. Minimum Inhibition Concentration (MIC)

Growth inhibition of *E. coli* was studied in a broth culture containing CPX or CPX-CS NPs. No antimicrobial activity was observed with bare chitosan NPs, even at concentration ranges between 0 and 100 μg/mL. The MIC value of CPX alone was 0.16 μg/mL, and that of CPX-CS NPs was 0.12–0.16 μg/mL. This result indicates that the antimicrobial activity of CPX-CS NPs was not compromised during the CPX-loaded nanoparticle manufacturing method. The MIC of CPX-loaded nanoparticles was comparable with CPX in solution and was not affected during the process of encapsulation into chitosan nanoparticles.

### 3.6. Cytotoxicity of CPX-Loaded Nanoparticles

In the preclinical development of a new pharmaceutical formulation, cytotoxicity characterization is one of the critical parameters. We studied different toxicity-induced parameters after exposure of the cells to nanoparticles, such as the trypan blue dye exclusion assay, neutral red uptake assay, and hemoglobin release assay for plasma membrane integrity of cells, viability of cells, and blood cell compatibility with the formulation.

In the trypan blue dye exclusion assay, we studied the effect of CPX-loaded NPs on the membrane integrity of the3T3 fibroblast cell line. Relative to the negative control, no statistically significant change was observed in the viability of cells in groups differently treated with blank NPs andCPX-loaded NPs (Figure 5A). In the neutral red uptake study, changes produced by toxic substances result in decreased uptake and binding of neutral red, making it possible to distinguish between viable, damaged, or dead cells via spectrophotometric measurements. Alterations of the cell surface or the sensitive lysosomal membrane lead to lysosomal fragility and other changes that gradually become irreversible. Cells treated with blank NPs or CPX-loaded NPs did not show any significant change in dye uptake with reference to the control. We examined the viability of 3T3 fibroblast cultures treated with different amounts of CPX in solution, CS-NPs, and CPX-CS NPs using the neutral red assay. The results showed that CPX in solution, CS-NPs, and CPX-CS NPs did not affect the viability of these cells at the doses used in the study. The cell viability plot (Figure 5C) showed that more than 91% of cells were viable after 24 h of incubation, even in the presence of very high concentrations of CS-NP and CPX-CS nanoparticles (1000 μg/mL). No cell morphology alteration of any cell lines was observed under the microscope. In addition, the cells were challenged up to 1 mg/mL of CS-NPs and CPX-CS NPs and their viability did not reduce, thus confirming no toxicity or minimal toxicity induced by CPX-loaded nanoparticles and the IC_50_ > 1 mg/mL. The results of the neutral red uptake assay did not indicate lysosomal fragility. The neutral red assay confirmed that CS-NP and CPX-CS NPs are non-toxic and are cytocompatible (Figure 2). Hemocompatibility is an important parameter in assessing the biological safety of nanomaterials. As a preliminary biocompatibility study of the formulation and their suitability as a formulation, we measured blood cell RBC compatibility by exposing CPX-loaded nanoparticles with RBCs (Figure 5B). No significant RBC lysis was observed after exposure to different concentrations of CS-NPs, CPX-CS NPs, and CPX in solution (less than 5% and comparable with the control).

### 3.7. Lipid Peroxidation for Membrane Damage Assay

Oxidative degradation of cell membranes after exposure to CPX-loaded nanoparticles was measured. The presence of reactive oxygen species (ROS) initiates membrane degradation. Oxidative degradation was measured after cell exposure to CPX-loaded nanoparticles using malondialdehyde or other thiobarbituricacid-reactive substances (Figure 6) [24]. It was shown that blank nanoparticles and CPX-loaded nanoparticles induced a dose-dependent peroxidative reaction of the PUFAs. In a normal cell comparison with the control group, lipid peroxidation was increased, and drug-containing nanoparticles could generate higher lipid peroxidation at concentrations of 100 µg/mL and above. However, the differences were not significant (*p* ≥ 0.05). The lipid peroxidation was higher than the drug in solution, which indirectly indicates the beneficial effect of drug delivery and efficient intracellular killing of bacteria in the infected host. Higher ROS generation plays an important role in drug delivery, as it is important in host defense against microbes and also suggests the activation of innate immunity.

## 4. Discussion

In dairy animals, mastitis may be caused by *E. coli* and *S. aureus*, which contaminate milk and reduce milk production [25,26]. It is very difficultto treat mastitis with antimicrobial treatments in dairyanimals due to milking at regular intervals, which interferes with maintaining the required concentration of antimicrobials at the site of infection. Antibiotic-loaded nanocarriers offer significant advantages over conventional therapy, such as targeted delivery tothe disease site and the controlled release of drugs, which help maintain optimal drug concentration at the affected site [27]. The present in vitro studies show CPX encapsulated into a chitosan-based nanocarrier was able to release the drug at physiological and acidic pH in a controlled manner, which can overcome the limitations of antimicrobial treatments of mastitis in dairy animals [28].

To overcome the challenges posed by maintaining a sufficient concentration of antimicrobials at the site of inflammation, we hypothesize that controlled release of antimicrobials from nanocarriers can address this problem. We encapsulated the broad-spectrum antimicrobial CPX into nano-size chitosan particles. The antimicrobial-encapsulated nano-size particles released CPX in acidic and basic pH media in a controlled manner; by 8 days it had released 60–70% of the total drug [28]. Stability of formulation plays a very important role and a critical factor in developing drug-loaded nanocarriers for therapeutic use. We studied the stability of chitosan nanoparticles synthesized using different ratios of chitosan and TPP and found that a 5:1 ratio was stable for upto 30 days in liquid formulation. We also observed that the drug release of CPX-loaded nanoparticles was pH-dependent. Theycan release their drugs faster at a higher pH than at alower pH (pH 7.4 > pH 5.2) [29]. The faster drug elution at neutral pH might be due to swelling of the polymer matrix, which is brought about by deprotonation of the amine group of chitosan. The drug release was increased with an increase in chitosan concentration [30,31]. The diffusion of the drug from the surface creates a pore in the matrix, which causes a channeling effect [30]. Incorporation of a higher concentration of drug causes more pore formation, leading to faster and higher drug release from the CPX-loaded nanoparticles, with a higher encapsulation efficiency. An uptake study of these particles shows it was easily uptaken by cells in invitro conditions. We also performed an efficacy study against clinical isolates of bacteria and CPX-loaded particles showed good efficacy in agar diffusion as well as in broth culture experiments. A study of theantibacterial activity of chitosan nanoparticles and ciprofloxacin-loaded nanoparticles against *E.coli* and *S. aureus* was conducted and found that the MIC of ciprofloxacin-loaded chitosan nanoparticles was 50% lower than that of ciprofloxacin hydrochloride alone in both microorganism species [32]. This study supports our findings that ciprofloxacin-loaded nanoparticles can reduce the dose of antimicrobials. In our cytotoxicity study, we used RBCs to incubate with CPX-loaded nanoparticles and found no or minimal toxicity compared to CPX alone. We performed different toxicity assays such as the MTT assay, hemolysis assay and lipid peroxidase assay. As prolonged tissue residues for most fluoroquinolones are not anticipated and determination of withdrawal times requires further study, we did not perform this study for our nanoparticles.

## 5. Conclusions

Our data show that CPX-loaded chitosan nanoparticles can provide a controlled drug release that depends on the surrounding tissue’s pH. CPX loaded into nanoparticles can kill bacteria efficiently. CPX-loaded chitosan nanoparticles can be used to effectively treat mastitis in dairy animals. Whileciprofloxacin has been listed by WHO as critically important microbial for human use, the present study proposes a strategy and provides a tool/platform which canbe used for development of nanoparticle-based formulations for the treatment of mastitis using fluroquinolones such as enrofloxacin that are approved for veterinary practice.

## Figures and Tables

**Figure 1 biomedicines-10-03282-f001:**
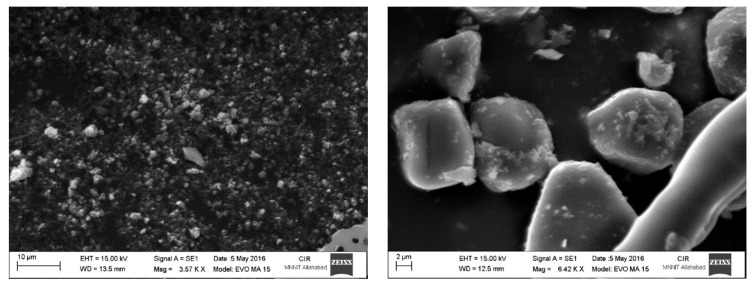
Surface morphology of CPX-loaded chitosan nanoparticles after lyophilization with trehalose cryoprotectant using scanning electron microscopy (SEM) image analysis.

**Figure 2 biomedicines-10-03282-f002:**
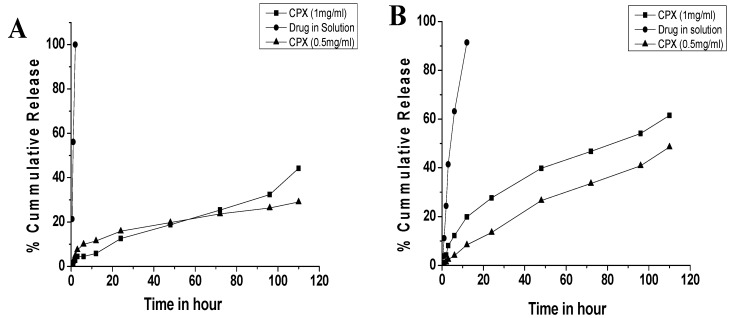
In vitro drug release study of ciprofloxacin hydrochloride-loaded chitosan nanoparticles in (**A**) PBS buffer at pH 5.2 and (**B**) PBS buffer at pH 7.4.

**Figure 3 biomedicines-10-03282-f003:**
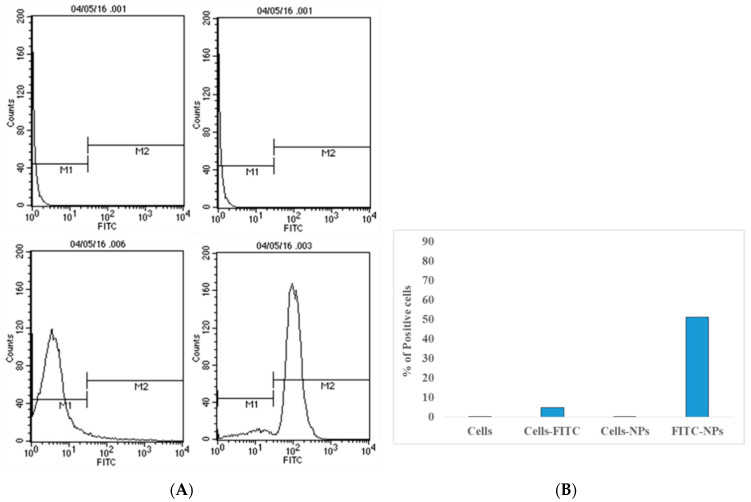
Cellular uptake of FITC dye-loaded chitosan nanoparticles (**A**) histogram of cells and (**B**) percentage of cell uptake of dye-loaded nanoparticles using flow cytometry analysis (*n* = 3, ±SD).

**Figure 4 biomedicines-10-03282-f004:**
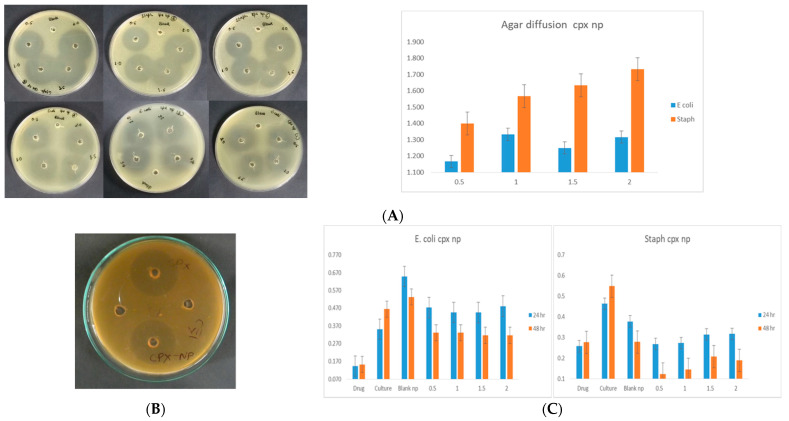
Antimicrobial activity of CPX-loaded nanoparticles against clinical isolate *S. aureus* and *E. coli*. (**A**) agar diffusion assay, (**B**) activity of equal amount of CPX and CPX-loaded nanoparticles, and (**C**) broth culture method at different doses of nanoparticles (*n* = 3, ±SD).

**Figure 5 biomedicines-10-03282-f005:**
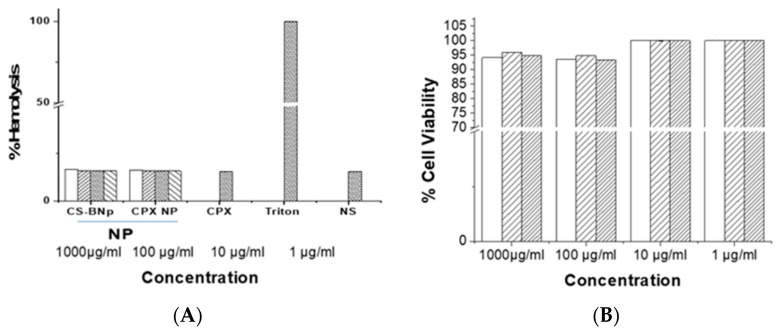
Toxicity assay after cells were exposed to CPX-loaded nanoparticles. (**A**) The trypan blue dye exclusion assay in 3T3 cells exposed todifferent doses of CPX, CPX-NPs, or blank NPs (1, 10, 100, and 1000 μg/mL); (**B**) hemolysis assay to assess membrane damage by CPX-loaded NPs; (**C**) neutral red dye uptake assay, cells exposed tociprofloxacin (CPX), chitosan blank nanoparticles (BNPs), and CPX-loaded chitosan nanoparticles (CPX-CS NPs) at different concentrations. Triton X-100 0.1% used as a positive control and normal saline (NS) used as negative control.

**Figure 6 biomedicines-10-03282-f006:**
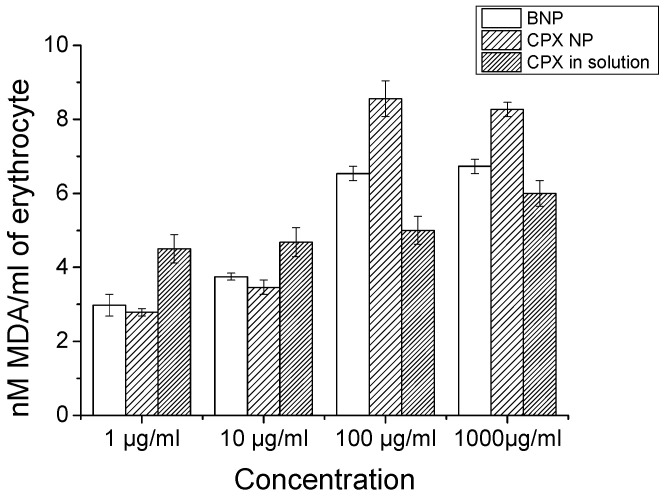
CPX-loaded chitosan nanoparticle exposure to the cells and membrane damage of cells assessed by lipid peroxidation assay (*n* = 3, ±SD).

**Table 1 biomedicines-10-03282-t001:** Encapsulation Efficiency, Particle Size, Polydispersity Index, and Zeta Potential of Ciprofloxacin (CPX)-Loaded Chitosan Nanoparticles (*n* = 5).

CS/TPP Weight Ratio	Ciprofloxacin Hydrochloride (mg/mL)	Average Particle Size (nm)	Polydispersity (PDI)	Zeta Potential (mV)	Encapsulation Efficiency (%)
5:1 (CS 1 mg/mL) & TPP (0.5 mg/mL) Formulation 1	0.5	195.6 ± 11	0.249	+24.86 ± 1.2	43 ± 2.29%
	1	185.5 ± 2	0.217	+24.91 ± 0.6	42 ± 2.35%
5:1 (CS 2 mg/mL) & PP (1 mg/mL) Formulation 2	0.5	214.8 ± 8	0.249	+29.6 ± 0.8	46 ± 2.15%
	1	252.9 ± 9	0.254	+27.0 ± 0.7	43 ± 2.27%
9:1 (CS 1 mg/mL) & TPP (1 mg/mL) Formulation 3	0.5	229.1 ± 13	0.250	+28.35 ± 2.6	43 ± 2.25%
	1	235.4 ± 16	0.289	+29.31 ± 2.1	43 ± 4.54%

**Table 2 biomedicines-10-03282-t002:** Particle Size, Polydispersity Index, and Drug Content of CPX-Loaded Chitosan Nanoparticles Stored at Room Temperature after 30 Days.

CS/TPP Weight Ratio	Average Particle Size (nm)	Polydispersity (PDI)	Drug Content
5:1 (CS 1 mg/mL & TPP) (0.5 mg/mL)	203.6 ± 22	0.269	96%
5:1 (CS 2 mg/mL & TPP) (1 mg/mL)	262.9 ± 35	0.284	96.2%
9:1 (CS 1 mg/mL & TPP) (1 mg/mL)	300.8 ± 31	0.350	85%

## Data Availability

Data are available with the authors.

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
