# Peer review of "Bioengineered Ciprofloxacin-Loaded Chitosan Nanoparticles for the Treatment of Bovine Mastitis"

_biomedicines, 2022, doi:10.3390/biomedicines10123282_

Round 1

Reviewer 1 Report

Dear Authors,

The present study is an in vivo research. The suggestion you declare are referred to in field or in vivo results. This is not acceptable and misleading. In vivo should be clearly mentioned in the title.

The information of the pathogens are vague..where they come from (clinical or subclinical cases of mastitis)

In conclusion the justification of the results with findings from others researches should be mentioned.

English syntax and grammar should be rechecked..

Author Response

Reviewer #1

Comments and Suggestions for Authors

Comment: The present study is an in vivo research. The suggestion you declare are referred to in field or in vivo results. This is not acceptable and misleading. In vivo should be clearly mentioned in the title.

Reply: Yes, the present paper content is only upto in-vitro level. Here we provided only in vitro results; we have not made any groups for in vivo study neither in the methodology nor in results.

Comment: The information of the pathogens are vague. where they come from (clinical or subclinical cases of mastitis).

Reply: We have explained the raised quarry in the Materials and Methods (Point 2.8).

Comment: In conclusion the justification of the results with findings from others researches should be mentioned.

Reply: We respect to the esteemed reviewer suggestions but in my knowledge that conclusion means authors should finally conclude the major finding of the study only. While supportive references are used in the discussion section. If still reviewers want then we can include. But usually references citations in the abstract and conclusion are not popular approach.

Comment: English syntax and grammar should be rechecked.

Reply: We are agreeing with the comments. Now we have critically corrected all syntax and grammar with the help of English speaking scientist.

Comment: Comments in the PDF file

Reply: We have considered all the suggestions and comments in the manuscript and accordingly corrected all in the revised manuscript.

Authors are grateful to the honorable reviewers for helping in improving the manuscript.

With deep regards

Dr. Neelesh Sharma

Division of Veterinary Medicine,

Faculty of Veterinary Sciences and Animal Husbandry, Sher-e-Kashmir University of

Agricultural Sciences & Technology of Jammu, R.S. Pura, Jammu, UT of Jammu &

Kashmir-181 102, India

Reviewer 2 Report

Dear authors

You describe a novel delivery system for intramammary formulations with an example antimicrobial.  The delivery system has a potential to be used in dairy animals in treatment of mastitis.

I sincerely hope my comments are taken as constructive criticism only.

I have attached all my comments in the PDF file.

Author Response

Reviewer #2

Comments:

You describe a novel delivery system for intramammary formulations with an example antimicrobial.  The delivery system has a potential to be used in dairy animals in treatment of mastitis.

I sincerely hope my comments are taken as constructive criticism only.

I have attached all my comments in the PDF file.

Reply: We have considered all the suggestions and comments in the manuscript and accordingly corrected all in the revised manuscript.

Authors are grateful to the honorable reviewers for helping in improving the manuscript.

With deep regards

Dr. Neelesh Sharma

Division of Veterinary Medicine,

Faculty of Veterinary Sciences and Animal Husbandry, Sher-e-Kashmir University of

Agricultural Sciences & Technology of Jammu, R.S. Pura, Jammu, UT of Jammu &

Kashmir-181 102, India

Round 2

Reviewer 1 Report

Dear Authors,

The majority of my suggestion were fulfilled. 

But I will insist on the title of the study.... A term declaring the potential prospects of the use of nanoparticles should be referred.

As I mentioned in previous comments in discussion a justification or a comparison with similar studies should be mentioned [Generalisability (external validity, applicability) of the trial findings..]

Author Response

The majority of my suggestion were fulfilled. 

Comment-1: But I will insist on the title of the study.... A term declaring the potential prospects of the use of nanoparticles should be referred.

Response: We are very sorry for missing your comment in the previous revision. Now we have modified the title as “Potential prospects of Bioengineered Ciprofloxacin-Loaded Chitosan Nanoparticles for the Treatment of Bovine Mastitis”

Comment-2: As I mentioned in previous comments in discussion a justification or a comparison with similar studies should be mentioned [Generalisability (external validity, applicability) of the trial findings..]

Response: We have mentioned a study [29] in the discussion section, which has also reported that stability is pH dependent.

We have also included a similar study [31]: A study on antibacterial activity of chitosan nanoparticles and ciprofloxacin-loaded nanoparticles against E.coli and S. aureus was conducted and found that MIC of ciprofloxacin loaded chitosan nanoparticles was 50% lower than that of ciprofloxacin hydrochloride alone in both of microorganism species [31]. This study supports the our findings that nano-particles loaded ciprofloxacin can reduce the dose of antimicrobials.

Reviewer 2 Report

Dear authors

I was happy to see your response that you have addressed all comments.  However, this was very short lived.  Reading the revised manuscript, I could see only 'cosmetic' improvements of the manuscript, but none of the important comments were addressed.

The following is essenital to be addressed:

1. WHO listing of ciprofloxacin (and all fluoroquinolones as critically-important antimicrobials that have to be restricted to treatment in humans not addressed.

2. The other important mechanisms known to affect the efficacy of intramammary treatments for S. aureus is the formation of biofilm.  Authors have not addressed this at all in the introduction.  This limitation should be addressed in the discussion section.  Recognition of limitation does not decrease the validity of the study.

3. Positive control needed.  Non-nano-Particularized ciprofloxacin should be used as positive control.   This needs repeating the experiment or alternatively (what I think will be more suitable) is a clear recognition of this limitation in the discussion.

4. Did authors considered the withholding period for the potential product?  There is need to strike a balance between efficacy and minimal withholding period of the formulation.  Even if this was not addressed, it is not detrimental to the study.  Has just to be acknowledged.

All my comments are included in the attached file

Author Response

The following is essential to be addressed:

Comment: WHO listing of ciprofloxacin (and all fluoroquinolones as critically-important antimicrobials that have to be restricted to treatment in humans not addressed.

Response:  I am thankful to the reviewer for raising this important point. We understand that WHO has listed ciprofloxaxin as critically important antimicrobial for human use. In Unites State of America, ciprofloxacin is prohibited from extralabel drug use (ELDU) in all food-producing animal species. Our purpose was to develop a proof of concept (POC) of by developing a nanoformulation using fluoroquinolones as a key drug for the treatment of mammary gland infection. Enrofloxacin, which is commonly prescribed in the veterinary medicine in Asian Countries including India, the metabolic conversion of enrofloxacin to ciprofloxacin was appreciable (36%). Therefore, the developed nanoformulation generates the proof of concept and provides a platform which can be used for the development of nanoparticles based treatment strategy for the treatment of mastitis using fluroquinolones approved for veterinary practices. At the same time, we should advocate prudent use of fluoroquinolones in clinical practice worldwide.

We have added a line in the conclusion section “Though ciproflocaxin has listed by WHO as critically important antimicrobial for human use, the present study propose a strategy and provide a tool/platform which can be used for development of nanoparticles based treatment strategy for the treatment of mastitis using fluroquinolones such as enrofloxacin approved for veterinary practice.

Moreover, ciprofloxacin is an “Off Lebel “drug mend for human use but is in frequent use in veterinary practice and very successful in treating bacterial infections, especially in cats and dogs. The most popular combination is marketed in the Ciprofloxacin-Tinidazole bolus. The reason for its frequent use in veterinary purposes in Indian and western countries is its low cost, readily available and high efficacy. Numerous publications are available in the public domain which shows the high efficacy of ciprofloxacin in animals.

Comment: The other important mechanisms known to affect the efficacy of intramammary treatments for S. aureus is the formation of biofilm.  Authors have not addressed this at all in the introduction.  This limitation should be addressed in the discussion section.  Recognition of limitation does not decrease the validity of the study.

Response: We have added the following line in introduction “Another challenge in the treatment of S. aureus infection is its ability to form biofilms, where bacteria in biofilms are more resistant to antibiotics compared to planktonic cells [7]. They are tightly packed in an extracellular polysaccharide matrix which helps them escape the immune response and antimicrobials in the environment [8]. This matrix hinders the penetration of many antibiotics, resulting in a significant decrease in antibiotic efficacy. Also, bacterial cells in deep layers of biofilm have a slow rate of metabolism and growth due to limited nutrient access [7]. NP-based drug delivery systems introduce a wide range of therapeutics, by either binding the drug to their large surface area or carrying it to the site of infection effectively, safely, and with a controlled rate of targeted delivery. They are capable of disrupting bacterial membranes and hindering biofilm formation, thus reducing the survival of the microorganism. A study by Thomas et. al. Evaluation of the efficacy of sustained release PLGA micro- and nanoparticles containing ciprofloxacin against bacterial biofilms demonstrated a sustained release over 6 days to effectively eradicate culturable S. aureus. The formulation might be a valuable alternative for the treatment of biofilms by achieving high local and sustained drug concentrations while minimizing systemic adverse effects and improving patient compliance [9].

Comment: Positive control needed.  Non-nano-Particularized ciprofloxacin should be used as positive control.   This needs repeating the experiment or alternatively (what I think will be more suitable) is a clear recognition of this limitation in the discussion.

Response: Control group has been added in the figure 4.

Comment: Did authors considered the withholding period for the potential product?  There is need to strike a balance between efficacy and minimal withholding period of the formulation.  Even if this was not addressed, it is not detrimental to the study.  Has just to be acknowledged.

Response: A text is assed in the discussion section “Although prolonged tissue residues for most of fluoroquinolones are not anticipated, the determination of withdrawal times requires further study.  Although prolonged tissue residues for most fluoroquinolones are not anticipated Ref: https://www.msdvetmanual.com/pharmacology/antibacterial-agents/quinolones,-including-fluoroquinolones,-use-in-animals

Comment: Broth (probably micro) dilution testing not included in the MM section yet but reported in the Abstract, results and discussion.

Response: We are highly thankful to the esteemed reviewer for catching the mistake. Now we have included microdilution method in the Material and Methods (Section 2.8).

Comment: Statistical methods not explained in MM section but later reported.

Response: We have not used any specific statistical model in the study. In methodology at suitable positions we have already been mention that experiment was performed in triplicate. Then its understood that we have to simply calculate the Mean-SD/SE

Comment: Reference for previous mastitis research on isolation and characterization of mastitis causing organisms.

Response: I am working on mastitis since many years and have been published many papers and 3 thesis. Few of them you can find below.

Huma Z.I., Sharma Neelesh, Kour S. and Lee S.J. (2022). Phenotypic and molecular characterization of bovine mastitis milk origin bacteria and linkage of intramammary infection with milk quality. Frontiers in Veterinary Science, 9: 885134.

Sharma Neelesh and Maiti S.K. (2010). Prevalence and etiology of subclinical mastitis in cows. The Indian Journal of Veterinary Research, 19(2):45-54.

Sharma Neelesh, Maiti S.K. and Sharma K.K. (2007). Prevalence, etiology and antibiogram of microorganisms associated with Sub-clinical mastitis in buffaloes in Durg, Chhattisgarh State (India). International Journal of Dairy Science, 2(2):145-151.

Sharma Neelesh, Pandey V. and Soodan J.S. (2010). Prevalence of mastitis in lactating dairy cows. Indian Journal of Veterinary Medicine, 30(2):102-104.

Round 3

Reviewer 2 Report

Thanks to the authors for addressing the issues raised in the previous review.

Just two notes

1. Authors again use antibiotic in the manuscript and their product is not an antibiotic, but it is an antmicrobial.  A simple find-replace search in Word will assist authors to change this omission.

2. The reviewe was not questioning the work on mastitis by the authors when asked for references.  The manuscript is self-standing.  Readers are not expected to search for literature published previously by the authors.  The references (2-3 at least) should be included where authors state that based on previous research on mastitis ....

Author Response

Comments: Authors again use antibiotic in the manuscript and their product is not an antibiotic, but it is an antmicrobial.  A simple find-replace search in Word will assist authors to change this omission.

Response: I have replaced remaining word “antibiotics” by “antimicrobials” in the manuscript.

Comment: The reviewe was not questioning the work on mastitis by the authors when asked for references.  The manuscript is self-standing.  Readers are not expected to search for literature published previously by the authors.  The references (2-3 at least) should be included where authors state that based on previous research on mastitis ....

Response: I have inserted my previous wok references in the revised manuscript.